# Initial and Delayed Metabolic Activity of Palatine Tonsils Measured with the PET/CT-Dedicated Parameters

**DOI:** 10.3390/diagnostics10100836

**Published:** 2020-10-17

**Authors:** Agata Pietrzak, Andrzej Marszalek, Malgorzata Paterska, Pawel Golusinski, Julitta Narozna, Witold Cholewinski

**Affiliations:** 1Electroradiology Department, Poznan University of Medical Sciences, Garbary 15, 61-866 Poznan, Poland; witoldc@onet.pl; 2Nuclear Medicine Department, Greater Poland Cancer Centre, Garbary 15, 61-866 Poznan, Poland; jnarozna@op.pl; 3Oncologic Pathology and Prophylaxis Department, Poznan University of Medical Sciences and the Greater Poland Cancer Center, Garbary 15, 61-866 Poznan, Poland; amars@ump.edu.pl; 4Pathology Department, Greater Poland Cancer Centre, Garbary 15, 61-866 Poznan, Poland; malgorzata.paterska@wco.pl; 5Department of Otolaryngology and Maxillofacial Surgery, University of Zielona Gora, 61-866 Poznan, Poland; pawel.golusinski@gmail.com; 6Poland and Biology and Environment Department, Poznan University of Medical Sciences, 61-866 Poznan, Poland

**Keywords:** head and neck squamous cell carcinoma, molecular diagnostics, palatine tonsils cancer, positron emission tomography, tonsillitis

## Abstract

One of the most critical elements in the palatine tonsils (PT) patients’ management is to distinguish chronic tonsillitis and malignant tumor. The single-time-point (STP) 2-deoxy-2-[^18^ F]fluoro-D-glucose positron emission tomography/computed tomography (^18^ F-FDG PET/CT) examination offers the most significant sensitivity and specificity in the head and neck (H&N) region evaluation among commonly used methods of imaging. However, introducing dual-time-point (DTP) scanning might improve the specificity and sensitivity of the technique, limited by the ^18^ F-FDG non-tumor-specific patterns, especially when comparing different metabolic parameters. The study aims to compare several surrogates of the maximal standardized uptake value (SUVmax), obtained in 36 subjects, divided into confirmed by pathologic study PT cancer and tonsillitis in patients who underwent DTP ^18^ F-FDG PET/CT scanning. In this study, we observed the increased sensitivity and the specificity of the DTP ^18^ F-FDG PET/CT when compared with the standard PET/CT protocol. It could be concluded that DTP ^18^ F-FDG PET/CT improves the PT cancer and chronic tonsillitis differential diagnosis.

## 1. Introduction

Palatine tonsil (PT) cancer is an aggressive and challenging to diagnose type of tumor in the head and neck (H&N) region. Most often, the possibility to detect the lesion within the PTs using the imaging method is limited due to insufficient spatial resolution of commonly used methods of imaging when compared with the tumor’s volume. Usually, the size of the primary tumor does not exceed a few millimeters (mm) while the mean spatial resolution of the diagnostic method approximates to one centimeter (cm). The most common histologic type of the PT malignancy is unilateral squamous cell carcinoma (SCC) and the diffuse large B-cell lymphoma (DLBCL). Sometimes, we can observe bilateral malignancy. However, in such cases, more possible is that the primary tumor has been diagnosed in advanced stage when it involves both sides of the oropharynx.

The imaging methods used in the H&N cancer patients management are the magnetic resonance imaging (MRI) and the single-time-point (STP) 2-deoxy-2 [^18^ F]fluoro-d-glucose positron emission tomography/computed tomography (^18^ F-FDG PET/CT) examination [1,2]. The ^18^ F-FDG PET/CT study provides the possibility to detect the metabolic abnormalities, which can be observed while the primary tumor is too small to be visible in vivo. However, additional delayed scanning protocol can improve the diagnostic accuracy of the PET/CT method due to increasing uptake of the glucose over time within the malignant tumor, measured with the retention index (RI) as a maximal standardized uptake value (SUVmax) surrogate, indicating the percentage change of the SUV value between the phases. In terms of the PT cancer, the most challenging seems to be detecting a small primary tumor and distinguishing between the tonsillitis and malignancy [3,4,5,6,7]. Very often, the tonsillitis caused by bacteria Actinomyces might be diagnosed incorrectly as high ^18^ F-FDG uptake caused by the developing neoplastic disease. Thus, using the additional scanning protocol seems to be of value. While the most commonly described dual-time-point (DTP) ^18^ F-FDG PET/CT protocols are performed at 60 and 120 min post-injection (p.i.) of the ^18^ F-FDG [5,6], authors [7,8] reported the possibility to perform the sequential DTP imaging as more convenient and easier to perform for differential diagnosis purposes. Moreover, evaluating various metabolic parameters and their surrogates using different scanning protocols might improve the diagnostic accuracy of the ^18^ F-FDG PET/CT method in H&N cancer patients [9,10,11].

Although, the delayed examinations in the H&N cancer and the RI-SUVmax value importance in the differential diagnosis have been reported previously, proportions and the differences between the obtained metabolic parameters have not been fully studied. The study aimed to evaluate several SUVmax derivatives reflecting initial and delayed glucose metabolism activity which might be useful in differentiation between inflammation and malignancy within PTs. The obtained data were supported by the histopathological diagnosis and the available immunohistochemical (IHC) studies.

## 2. Materials & Methods

### 2.1. Bioethics

This study was designed per received of the patients’ written informed consent and approved by the Local Bioethical Committee (Poznan University of Medical Sciences Bioethical Committee; Chair: prof. Pawel Checinski) as the retrospective analysis based on standardly performed procedures, conducted between January 2015 to January 2020 (study approved: 14.01.2019). To ensure the investigations’ transparency, some of the raw data have been included into the article’s content. All data have been strictly anonymized.

### 2.2. Database, Software, and Measured Parameters

In this study, we analyzed 36 patients (7 women, 29 men) diagnosed with tonsillitis, DLBCL and SCC. All studied subjects underwent the contrast-enhanced computed tomography (ceCT) before the PET/CT. The ceCT scanning resulted in detecting suspicious cervical lymph nodes. The indication for the PET/CT examination was the cancer of unknown primary (CUP syndrome).

We divided the studied subjects into the following groups considering the histopathologic diagnosis: inflammation (tonsillitis, I; 18 patients), malignancy (SCC and DLBCL; M;18 cases). The inclusion criteria were the unilateral PT malignancy, delayed (DTP) ^18^ F-FDG PET/CT scanning performed, histopathologic diagnosis available, no treatment applied to the patient before the scanning. The exclusion criteria were: STP protocol only, no histopathologic confirmation available, surgical resection or other therapy introduced before the acquisition. We included in observations the available IHC measurements. All of the analyzed patients underwent the DTP ^18^ F-FDG PET/CT examinations before the treatment to assess the CUP syndrome or to establish the stage of the H&N cancer prior therapy using similar scanning protocol (Table 1).

We used the Statistica 13.0 (Statsoft, Poland; available upon the individual license) application to perform the necessary analyses. We used the MiM 7.0 (MiM Software Inc. Cleveland, Ohio, USA); no commercial license available, (Figure 1 and Figure 2) software for contouring and SUVmax surrogates’ calculation. We have used the Philips Gemini TF 16 (Ohio, Cleveland, USA) PET/CT scanner in all patients.

### 2.3. Measured Parameters—Overview and Descriptions

We measured the following parameters: SUVmax within the normal (N) PT (SUVmaxN) and lesion (L; SUVmaxI, M—inflammation, I or malignancy, M), the absolute difference between the SUVmax within lesion and normal contralateral PT (SUVmaxL–SUVmaxN), and the SUVmaxL and SUVmaxN proportion in both phases of scanning. We evaluated the SUVmax90 min/SUVmax60 min proportions within the normal and pathologic areas as well. We obtained the RI-SUVmax in normal and abnormal PTs.

In this study, we used the following indicators: N—normal PT (no metabolic abnormalities), L—lesion (inflammation—I, malignancy—M). When described the results and used the following descriptions: group I (patients in whom we found tonsillitis within the one of PTs), group II (confirmed unilateral PT malignancy). Most of the time, we used the term inflammation instead of malignancy to directly indicate the benign and non-benign finding. To evaluate the differences between the SUVmax within the lesion and normal structure in each patient, SUVmax proportions, RI-SUVmax, we used the following equations, respectively:**Difference_L−N_** = SUVmax_L_ − SUVmax_N_ (both in I and II phase of scanning),**Proportion_L/N_** = SUVmax_L_/SUVmax_N_ (both in early and delayed scanning),**Proportion_90/60_** = SUVmaxL_90_/SUVmaxL_60_ and SUVmaxN_90_/SUVmaxN_60_,**RI-SUVmax** = (SUVmax_90_ − SUVmax_60_)/SUVmax_60_ (both for N, L).

### 2.4. Histopathologic and IHC Data

We used the following parameters: histologic type of the lesion, specific IHC parameters (Table 2).

In all cases, PT inflammation was caused by the bacteria Actinomyces. Within the group of PT malignant tumors, we found that most of them were SCC (keratotic and aceratotic, grade G2, G3; 14 patients) and in 4 cases, the histopathologic examination indicated the DLBCL (in young male patients). In most of the examined patients, HPV infection was investigated using the p16 marker.

The limitation of the method was the unavailability of the IHC data in 6 patients (5—PT inflammation, 1—PT tumor). In some cases, these studies were not applied due to specific reasons (insufficient or unavailable to examine material—biopsy performed in the different institution) or because of no indications to perform IHC measurements.

## 3. Results

### 3.1. Statistics

We performed the commonly mentioned in the biomedical literature statistical analyses. We used the statistical significance level α = 0.05 (confidence interval at the level of 95%, CI_95_), following the Shapiro–Wilk test’s results when the variables’ normality was analyzed. We used statistical tests available for small samples and two-tailed models to ensure the lack of investigators’ predictions influence in terms of the results of presented calculations. Based on previously published DTP studies [7,8], the metabolic activity differences between different types of lesions might be difficult to assess when using natural numbers describing SUVmax value level. Thus, to ensure the detailed analysis and minimize the calculation errors, we used two decimals when measuring the SUVmax values. During the analyses, we tested null and alternative hypotheses (H_0,_ H_a_) that H_0_ suggests no significant change of the metabolic activity over time and no differences between compared groups. H_a_ assumed two-tailed possibility of change observed in our samples. Based on the previously published results [7,8], we might expect a significant decrease of metabolic activity within normal PT, no change within benign lesions and increasing over time SUVmax value within malignant tumors. However, we assumed that randomized sampling might result in different clinical conclusions. Thus, we considered two-tailed test as the most reliable despite decreased power of the statistical test. We interpreted the statistical test, considering p-value. When *p* < 0.05, we rejected H_0_ (statistically significant differences observed) and when *p* ≥ 0.05, the H_0_ was not rejected (we assumed that there is a possibility of no differences between compared groups). We indicated the type of the test in an appropriate section describing each measurement.

The studied group was homogenous in terms of age, thus, we did not perform the analysis of correlation between the age and PT pathologies occurrence. The mean age ± standard deviation (S.D.) of examined subjects was 60 ± 11 years old (y.o.), range: 24 to 75 y.o.

We studied 20 lesions in the left PT and 16—right PT. We compared the histopathologic diagnosis with the nuclear medicine specialists’ suggestions based on the PET/CT images interpretation to obtain the ability of the method to evaluate the type of lesion. The analysis showed 3 false-positive (PT inflammation suggested to be the tumor) and 2 false-negative results (2 DLCBL lesions assessed as possibly chronic tonsillitis). The number of false-positive and false-negative effects occurred as statistically insignificant with *p* = 0.18 (Fisher’s exact test’s).

### 3.2. SUVmax at 60 and 90 min. p.i.

We evaluated the SUVmax values, obtained within PTs in 36 patients. In each patient, we compared the normal and abnormal PT glucose metabolism activity, measured with the SUVmax and obtained in initial and delayed PET/CT phase of scanning. We evaluated 72 PTs (36 abnormal: I, M, 36—N). We obtained the SUVmax value within PTs, the absolute difference between the normal and abnormal PTs, and the SUVmax value proportions.

Table 3 shows the SUVmax value data, obtained in group I and II during the initial (60 min p.i.) and delayed (90 min p.i.) scanning:

### 3.3. SUVmax Value Changes over Time and the RI-SUVmax

We evaluated the SUVmax values at 60 and 90 min p.i. of the ^18^ F-FDG dataset’s distribution, using the following samples: malignant lesions (18 cases; *p* = 0.01, 0.01, respectively), inflammation/tonsillitis (18 cases; *p* = 0.08, 0.09, respectively), all abnormal lesions (36 cases; *p* < 0.001, <0.001, respectively), normal PT in group I (18 cases; *p* = 0.06, 0.06, respectively), normal PT in group II (18 cases; *p* = 0.16, 0.88, respectively), all normal PTs (36 cases; *p* = 0.06, 0.09, respectively). SUVmax values datasets were normally distributed within normal and inflammation PTs groups. The variables’ distribution significantly differed from Gaussian when the malignant and all abnormal lesions group was analyzed.

We assessed the SUVmax values changes over time between the phases. In all cases, we compared two dependent groups of variables (same group of patients, evaluated in two-time points—initial and delayed phase of scanning). To compare the normally distributed variables, we used *t*-test for dependent variables, in other cases—Wilcoxon’s pair test. To study the normal and abnormal metabolic activity, we used the histologic type of a lesion as the independent variable in the Mann–Whitney’s U test. Table 4 shows the results obtained in our sample:

When we analyzed the STP ^18^ F-FDG PET/CT scanning, we found no significant differences between normal metabolic activity and PT inflammation. The difference between inflammation and malignant lesions was significant with *p* = 0.04. According to literature [4,8], the SUVmax value in normal structures can be obtained as similar or significantly lower on initial and delayed scans when using DTP ^18^ F-FDG PET/CT study protocol. As expected, we found the normal PTs comparable in both groups of patients.

Another indicator describing the SUVmax value changes over time, used for the delayed PET/CT studies evaluation is the RI-SUVmax [2,4]. In this sample, the RI-SUVmax ± S.D. within the malignant lesions, inflammation, and normal PTs were, respectively: 12 ± 14%, 8 ± 13%, −4 ± 23%. According to the Mann–Whitney’s U test’s results, the RI-SUVmax differed significantly between the PT tumor and inflammation with *p* < 0.001. The normal and abnormal (inflammation, malignancy) metabolic activity differed significantly with *p* < 0.001.

### 3.4. Absolute SUVmax Value Difference and Proportions

To compare the groups of analysis and to find differences between measurements, which might be considered helpful in terms of distinguishing inflammation and malignancy within PTs, we used the *t*-test, Mann–Whitney’s U test, and the ANOVA. The differences and proportions of SUVmax value within normal PTs at 60 and at 90 min p.i. were insignificant in both examined groups. Table 5 shows detailed characteristics:

The glucose metabolism activity decreased over time within the normal PTs and increased within the inflammation and PT tumor. We analyzed the obtained results to find the possible predictor, which might be considered helpful in terms of distinguishing the inflammatory and malignant tumors of PT using the DTP ^18^ F-FDG PET/CT studies.

### 3.5. SUVmax Value Differences

When comparing the differences obtained at 60, 90 min p.i. between the lesion (inflammation or malignant tumor) and normal PT, we found the differences significant between the group I and group II with *p* = 0.02 at 60 min and *p* < 0.001 at 90 min p.i. Because the difference was greater after delayed scanning, we performed the receiver operating characteristics curves (ROC) analysis to find the predictive SUVmax value difference between the lesion and normal PT which might distinguish between the inflammation and malignant tumor (Figure 3):

We did not compared the cut-off point differentiating the inflammation and PT tumor based on the differences between the normal and abnormal PTs as a demanding additional factor, indicating the relative SUVmax value difference, obtained at 60 and 90 min p.i. Providing this factor would make the analysis confusing and possibly less reliable due to the necessity of using mean values on every step of the analysis, increasing the standard error in the sample.

The analysis of the SUVmax value differences (SUVmax90–SUVmax60) within the lesion alone significantly differed between the inflammation and the PT tumor with *p* = 0.02. Thus, the SUVmax difference cut-off evaluated with the ROC analysis, which differentiates inflammation and malignancy within PT, might be considered helpful. We observed decreased sensitivity and specificity of the DTP study when compared with the analysis regarding the normal PTs metabolic activity (Figure 4):

### 3.6. SUVmax Value Proportions

When comparing the proportions indicating the SUVmax value within the lesion and normal PT, we found that the ratios significantly differed between the initial and delayed phase of scanning with *p* < 0.001 in both groups. However, the essential step of the analysis was to indicate the differences between the inflammation and the malignant PT tumor. To examine that, we compared the differences and proportions between the inflammatory and malignant lesions using the ANOVA test. We evaluated the differences and proportions of the SUVmax value at 60, 90 min p.i. of the ^18^ F-FDG within the lesion and normal PT in both groups, using the type of the lesion as the qualitative (independent) variable. When studied the SUVmax_I/N_ and SUVmax_M/N,_ we found strong difference between results obtained in group I and II (*p* < 0.001).

According to the ROC analysis results, the SUVmax proportion cut-off between the lesion and normal PT SUVmax, which differentiates inflammation and malignant PT tumor, was 1.53 at 60 min p.i. (sensitivity/specificity: 72/94%, Youden Index = 0.67) and at 90 min p.i.—1.92 (sensitivity/specificity: 67/100%, Youden Index = 0.67). When comparing the SUVmax proportion of lesions metabolic activity at 90 vs. 60 min p.i., we found the cut-off value of 1.12 (sensitivity/specificity: 50/72%, Youden Index = 0.28, Figure 5).

In this study, including the measurements based on the normal and abnormal metabolic activity, increased the sensitivity and the specificity of the PET/CT method. When comparing the SUVmax proportions within the lesion only, the diagnosis seemed to be less precise.

### 3.7. SUVmax Cut-Off Value at 60, 90 min p.i.

According to the literature [9], the SUVmax value exceeding 2.50 may suggest abnormality within the ROI. In terms of PTs, the SUVmax value at the level 2.50 indicates most often the physiologic glucose metabolism level. In our sample, the SUVmax value within normal PTs ranged between 2.58 and 3.98 initially and 2.52 to 3.93 in the delayed phase of scanning, which makes the 2.50 an implausible value according to at least 95% of the examined population.

The SUVmax value within the PT inflammation on the initial and delayed PET/CT scans ranged, respectively, from 3.21 to 4.79, 3.43 to 5.18. The PT tumors SUVmax value was 5.45 to 9.92 at 60 min p.i. and 6.02 to 10.75 at 90 min p.i. of the ^18^ F-FDG. The SUVmax value cut-off points at 60 and 90 min p.i. of the ^18^ F-FDG were: 4.10 (sensitivity/specificity = 83/67%, Youden Index = 0.50), 4.44 (sensitivity/specificity: 85%/71%, Youden Index = 0.50), respectively.

## 4. Discussion

According to the Warburg effect [12], malignant lesions utilize glucose significantly higher when compared with the normal structures due to increased metabolism of the cancer cells. The main limitation of the ^18^ F-FDG PET/CT study is the non tumor-specific pattern of the radiopharmaceutical ^18^ F-FDG, which decreases the specificity of the method in distinguishing a malignant tumor and metabolically active benign lesion (i.e., inflammation). In this study, we obtained several SUVmax value surrogates to evaluate the usefulness of the DTP ^18^ F-FDG PET/CT examination in distinguishing PT tumor and chronic inflammation, which can be considered difficult to differentiate using the standard STP ^18^ F-FDG PET/CT scanning protocol.

According to our results, the SUVmax value change over time calculation was not helpful in terms of PT tumor and inflammation differential diagnosis due to significant increase of the SUVmax value on delayed scans in both benign and malignant lesions. To find the differences between the analyzed groups of lesions, we performed more complex statistical analyses. In this study, the results might be considered sample-size-dependent. It was previously reported [7,8] that including more numerous groups of cases has shown no significant changes on initial and delayed scans within benign lesions. At the same time, the ^18^ F-FDG uptake increase within malignant tumors was relevant. This suggests that to ensure the DTP ^18^ F-FDG PET/CT study usefulness regardless of the sample-size and specific diagnosis, obtaining several metabolic activity indicators might be helpful.

In this sample, the RI-SUVmax ± S.D. within the malignant lesions, inflammation, and normal PTs were, respectively, 12 ± 14%, 8 ± 13%, −4 ± 23%. The RI-SUVmax is a widely described, delayed protocols-dedicated parameter. As expected, we observed a high increase of the SUVmax value within the malignant lesions, medium within the inflammation, and decreasing tendency within the normal PTs. When comparing three groups of PTs, we observed statistical difference (*p* < 0.001) between the groups, especially when analyzing the difference between all normal and all abnormal PTs. In this study, the SUVmax value cut-off, which differentiated between benign and malignant lesions’ metabolic activity, was 4.10 on initial and 4.44 on delayed images. However, the DTP ^18^ F-FDG PET/CT study increased the sensitivity and specificity of the method only by 2% and 4%, respectively.

The novelty in the DTP ^18^ F-FDG PET/CT studies analysis might be evaluating the absolute difference between the SUVmax value obtained within the lesion and normal PT in the same patients and calculating specific SUVmax value proportions. In our study, the absolute difference of SUVmax_L_−SUVmax_N_ in the inflammation patients’ sample (group I), and cancer patients’ dataset (group II), differed significantly, especially after delayed scanning. According to the ROC analysis results, the SUVmax value cut-off, which differentiates between the inflammation and malignant PT tumor in our sample, was 2.19 with the sensitivity up to 72% and the specificity—89% when using the DTP ^18^ F-FDG PET/CT study. The SUVmax proportions differed when we obtained the lesions and normal PTs vs proportions calculated within the lesions alone. The SUVmax proportion between lesions and normal PTs equaled 1.53 at 60 and 1.92 at 90 min p.i. with specificity of the method up to 100% in the delayed phase of scanning. When comparing the results with those obtained within the lesions alone, we found the cut-off value at the level of 1.12 with significantly lower sensitivity and specificity.

Based on the ROC analysis results, we observed that the sensitivity and specificity of the DTP ^18^ F-FDG PET/CT method are higher when compared the lesion and normal PT instead of metabolic activity changes within the lesion alone. It leads to the conclusion that obtaining the contralateral normal PT in patients suspected of malignancy, might help in distinguishing inflammation and malignant tumor with the sensitivity and specificity up to 72%, 100%, respectively. In our study, the DTP ^18^ F-FDG PET/CT increased the overall sensitivity and specificity of the method in terms of the inflammation and malignant PT tumor differential diagnosis. However, we could not compare the results with the literature due to lack of similar research.

## 5. Conclusions

The study showed that in the case of doubt regarding the type of observed lesion within the PT, calculating differences and proportions between the observed lesion and contralateral normal PT might be considered helpful. The appropriate indication is highly valuable when the histopathologic examination is considered, and there is a necessity to suggest the areas which should undergo biopsy. It seems to be even more important in the case of patients in whom we observe multiple areas of increased metabolic activity. When using the DTP studies, the possibility to appropriately indicate the area which should undergo further examination seems to be more convenient.

## Figures and Tables

**Figure 1 diagnostics-10-00836-f001:**
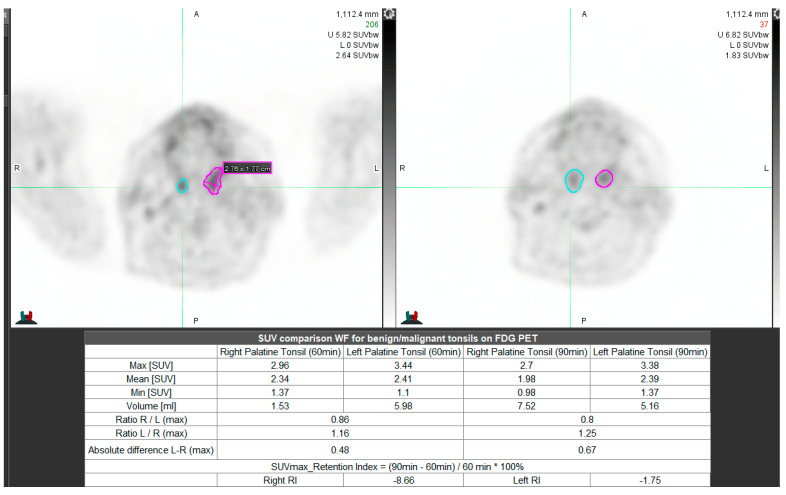
Unilateral inflammation (MiM 7.0 software). Description: Left upper image—initial, right upper scan—delayed scanning.

**Figure 2 diagnostics-10-00836-f002:**
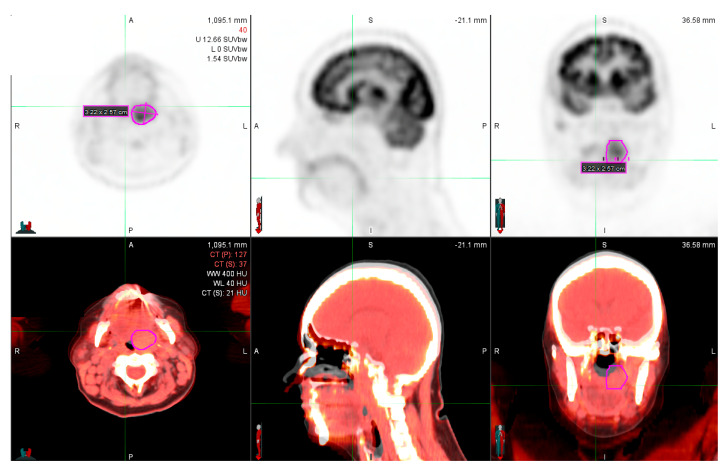
Left palatine tonsil primary tumor (MiM 7.0 fusion) obtained in one patient – initial scanning at 60min.p.i. (upper images: PET, below—PET/CT fusion).

**Figure 3 diagnostics-10-00836-f003:**
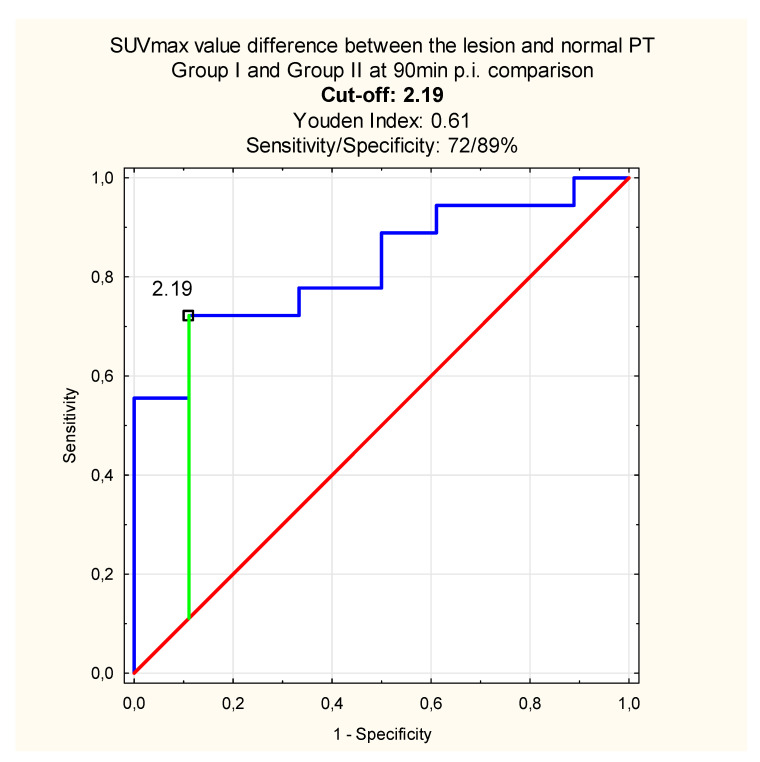
The SUVmax value difference between the lesion and normal PT which distinguishes inflammation and PT tumor at 90 min p.i. (blue line—sensitivity, red line—specificity, green—cut-off point).

**Figure 4 diagnostics-10-00836-f004:**
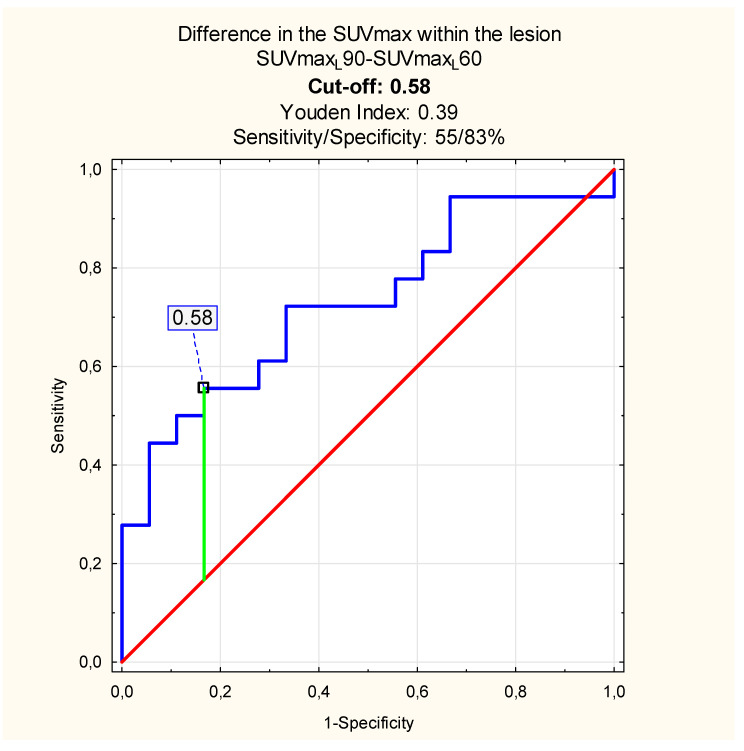
SUVmax max value cut-off: SUVmax within the lesion at 90 vs. 60 min p.i. (blue line—sensitivity, red line—specificity, green—cut-off point).

**Figure 5 diagnostics-10-00836-f005:**
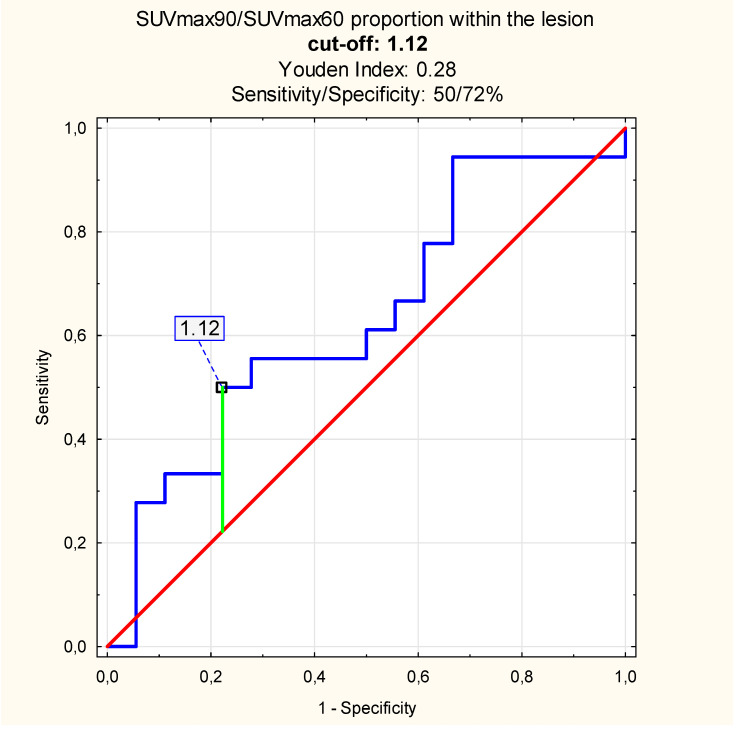
The proportion between the SUVmax value at 90, 60 min p.i. within the inflammation and malignant PT tumor: cut-off value (blue line—sensitivity, red line—specificity, green—cut-off point).

**Table 1 diagnostics-10-00836-t001:** Scanning protocol.

Phases/Area of Interest	Schedule	Value [min]
**Initial: 60 min p.i.**	mean start time p.i. ± S.D.	57 ± 2
Whole body scanning: mid-thigh–skull vertex	Range p.i.	55–64
**Delayed: 90 min p.i.**	mean start time p.i. ± S.D.	88 ± 3
Head and neck region: skull vertex–aortic arch	Range p.i.	79–92
Initial and delayed	mean total delay between phases	5 ± 3
	Range	5–8

Abbreviations: min—minutes, p.i.—post injection, S.D.—standard deviation.

**Table 2 diagnostics-10-00836-t002:** IHC markers obtained in our patients.

Parameter	Description—Indication
AE_1,2,3_	Anticytokeratin monoclonal antibodies; SCC diagnosis
Bcl_2,6_	Apoptotic processes proteins; b-cell lymphoma 2 and 6
CD_3,4,8,10_	Cluster of differentiation proteins; lymphoma diagnosis
CK_7_	Cytokeratin 7, sarcolectin; type II keratin; SCC diagnosis
D_1_	Cyklin; lymphoma
Ki-67	Antibody, proliferation marker; SCC
MUM_1_	Transcriptional factor, i.e., lymphoma diagnosis
p16	Tumor suppressor protein, i.e., HPV detection
p63	Monoclonal mouse anti-human tumor protein; SCC

Abbreviations: 1,2,3…—subtypes of markers, SCC—squamous cell carcinoma, CD—cluster for differentiation protein, CK—cytokeratin, —and other; Ki-67, MUM,D—protein markers, HPV—human papilloma virus.

**Table 3 diagnostics-10-00836-t003:** The SUVmax values at 60 and 90 min p.i. of the ^18^ F-FDG.

Parameter/ROI*	Avg SUVmax ± S.D.	Me	CI_95_ [−…; + …]
**Group I**			
N_60_	3.23 ± 1.30	2.83	2.58;3.88
N_90_	3.22 ± 1.40	2.74	2.52;3.91
I_60_	4.00 ± 1.58	3.52	3.21;4.79
I_90_	4.30 ± 1.76	3.85	3.43;5.18
**Group II**			
N_60_	3.44 ± 1.08	3.27	2.91;3.98
N_90_	3.30 ± 1.27	3.08	2.66;3.93
M_60_	7.69 ± 4.49	5.13	5.45;9.92
M_90_	8.38 ± 4.76	6.57	6.02;10.75

Abbreviation: SUVmax—maximal standardized value, ^18^ F-FDG—fluorine-^18^ F-fluorodeoxyglucose, CI—confidence interval, N—normal palatine tonsils, I—inflammation (tonsillitis), M—malignant lesions, ROI*—region of interest (N or I, M PT), Me—median.

**Table 4 diagnostics-10-00836-t004:** The SUVmax changes over time.

Group	*p*-Value	Conclusion (SUVmax Change over Time)
**Changes over Time**		
Malignant lesions	<0.001	significant increase
Inflammation	0.01	significant increase
Normal PT in group I *	0.83	No change
Normal PT in group II *	0.41	No change
**Differences between two groups of analysis**		
All abnormal lesions	<0.001	**Significantly different**
All normal PTs	0.74	Comparable

Abbreviations: SUVmax—maximal standardized uptake value, PT—palatine tonsil, *—group I: normal PTs vs. inflammation, group II: normal PTs vs. malignant PT tumor.

**Table 5 diagnostics-10-00836-t005:** Normal and abnormal PT comparison: differences and proportions.

Parameter	Value				
	Avg ± S.D.	Me	CI_95:_ [−…;… + ]	Range	*p*-Value (Distribution)
**Group I (inflammation and normal PT)**					
SUVmax60 _I-N_	0.77 ± 0.57	0.64	0.49;1.06	0.07–2.01	0.06
SUVmax90 _I-N_	1.09 ± 0.71	0.90	0.73;1.44	0.03–2.46	0.39
SUVmax60 _I_/SUVmax60 _N_	1.25 ± 0.18	1.20	1.16;1.34	0.97–1.73	0.28
SUVmax90 _I_/SUVmax90 _N_	1.36 ± 0.21	1.32	1.26;1.46	1.01–1.74	0.59
SUVmax90 _I_/SUVmax60 _L_	1.08 ± 0.13	1.09	1.01;1.14	0.84–1.46	0.07
**Group II (malignant PT tumor and normal PT)**					
SUVmax60 _M-N_	4.25 ± 4.32	2.13	2.10;6.40	0.21–15.65	0.007
SUVmax90 _M-N_	5.09 ± 4.84	2.74	2.68;7.49	0.22–18.74	0.007
SUVmax60 _M_/SUVmax60 _N_	2.31 ± 1.44	1.92	1.59;3.03	1.04–6.65	<0.001
SUVmax90 _M_/SUVmax90 _N_	3.22 ± 3.78	2.01	1.34;5.10	1.06–17.58	<0.001
SUVmax90 _M_/SUVmax60 _L_	1.11 ± 0.14	1.11	1.04;1.18	0.69–1.35	0.06
**SUVmax difference within the lesion (SUVmax90-SUVmax60)**					
Inflammation	0.30 ± 0.46	0.33	0.07;0.53	−0.40–1.51	0.36
PT tumor	0.70 ± 1.36	0.68	0.02;1.37	−3.95–2.32	<0.001

Abbreviations: PT—palatine tonsil, avg—average, S.D.—standard deviation, Me—median, CT—confidence interval, SUVmax—maximal standardized uptake value, I—inflammation, M—malignant tumor, N—normal P.

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
