# Peer review of "Initial and Delayed Metabolic Activity of Palatine Tonsils Measured with the PET/CT-Dedicated Parameters"

_diagnostics, 2020, doi:10.3390/diagnostics10100836_

Round 1
Reviewer 1 Report
The study is well conduct as well as written (language needs very minor correction, please revised). Authors are not new into the topic Pietrzak, A.K., Marszalek, A., Kazmierska, J. et al. Sequential delayed [18 F]FDG PET/CT examinations in the pharynx. Sci Rep 10, 2910 (2020). https://doi.org/10.1038/s41598-020-59832-4.
Despite being reported in the discussion section, this article appears to be an extension of the same study carried out on the pharynx. Please authors may comment on this my issue?
Author Response
We would like to sincerely thank the Reviewer #1 for kind and valuable remarks.
The study is well conduct as well as written (language needs very minor correction, please revised).
Thank you, the article’s language has been spell-checked and adjusted.
Authors are not new into the topic Pietrzak, A.K., Marszalek, A., Kazmierska, J. et al. Sequential delayed [18 F]FDG PET/CT examinations in the pharynx. Sci Rep 10, 2910 (2020). https://doi.org/10.1038/s41598-020-59832-4. Despite being reported in the discussion section, this article appears to be an extension of the same study carried out on the pharynx. Please authors may comment on this my issue?
Thank you for the remark. Previous articles considering the usefulness of the DTP 18F-FDG PET/CT studies in differential diagnosis have been conducted in terms of SUVmax value cut-off points using ROC curves and comparing normal and abnormal metabolic activity in the region of interest (different than presented in this study). We limited the statistical evaluation to mean values and cut-offs, however we did not consider the proportions and differences between abnormality found in one tonsil and contralateral normal organ. It was mainly because we needed to study paired structures as palatine tonsils (PTs).
Moreover, we have been interested whether the proportions and differences calculations are meaningful and does it make it more significant when using standard single-time-point scanning or dual-time-point. We have found that the DTP studies really may improve the diagnosis and differences/proportions could be used when the cut-off values are not helpful. We could not compare the results with other authors, because – according to our knowledge – such study has not been yet published elsewhere. When we analyzed published papers on comparable issue, it was clear that the obtained values occurred significantly different when comparing tonsillitis and palatine tonsil cancer. We have also discovered for the first time, that evaluating normal metabolic activity (including pairs: normal PT and abnormal PT) make a difference in the overall analysis. Previously, referential normal metabolic activity was not always helpful.
Reviewer 2 Report
Material and Methods
Line 104 to 108: I recommend not to use results in material and methods section. If the authors did not find any correlation between age and sex, please include it within results. Moreover, I do not understand completely "the comparison has been neglected". It is the same with the age, please I recommend using the values in results.
Line 143: Why did they decide to set the group in the same category SCC and DLBCL? Do the malignant lesions have the same characteristics when they are evaluated in PET/CT and by histopathological studies? I recommend, as possible, classifying according to histopathological results and then make the comparison by PET/CT.
Results
Line 268: What do they mean with “normal lesion”?
Line 269: “We have…” I consider that this paragraph is more suitable for the conclusions section.
Discussion
The discussion and conclusion were well described and provide new information that may help to differentiate between benign and malignant when PET/CT is used.
Overall comments:
This study was well made, the authors compared tonsillar lesions and results with the PET/CT. I think that authors provide interesting information about differential diagnosis using PET/CT (I, vs M). However, I have some questions that authors need to resolve.
1. Why did not the authors use any photomicrographs of the carcinomas and lymphomas, as well as photomicrographs of the immunohistochemistry?
2. Why did the authors encompass only one group of carcinomas and lymphomas, when the histopathological features and behavior are different?
3. I think it may be interesting to only contrast malignant lesions (SCC vs, DLBCL) and PET/CT. The results could be different when these pathologies are compared with inflammatory PT.
Author Response
We would like to sincerely thank Reviewer #2 for extended comments. We found them very helpful and we have adjusted paper accordingly with Referee #2 remarks.
Material and Methods: Line 104 to 108: I recommend not to use results in the material and methods section. If the authors did not find any correlation between age and sex, please include it within results. Moreover, I do not understand completely "the comparison has been neglected". It is the same with the age, please I recommend using the values in results.
Thank you for the comment. We have recorrected the description and placed the information in the Results section. We have also clarified: because the group was homogenous in terms of age, we did not perform the correlation analysis. It was expected to obtain a very high positive correlation in this database, which is not reliable in this database.
Lines: 102-106 and 194-197.
Line 143: Why did they decide to set the group in the same category SCC and DLBCL? Do the malignant lesions have the same characteristics when they are evaluated in PET/CT and by histopathological studies? I recommend, as possible, classifying according to histopathological results and then make the comparison by PET/CT.
Thank you for the comment. In fact, when the disease is not advanced and the only symptom of the possible illness is the presence of metastatic cervical lymph nodes (we have clarified this matter in the methods section after the review), the palatine tonsils SCC and DLBCL might be difficult to differentiate. We have studied patients with CUP syndrome, so after extended diagnosis, oncologists concluded that it might be SCC or DLBCL. Every patient included into the groups underwent contrast-enhanced CT before the PET and because of CUP-syndrome (suspicious lymph nodes, no primary tumour obtainable using ceCT), patients’ have been studied with the PET/CT.
We have included into submission the Supplementary Materials: two extended tables in which we have included the IHC and histopathologic diagnosis with the results obtained with the PET/CT: final conclusion indicating the possible type of a lesion.
Results
Line 268: What do they mean with “normal lesion”?
Thank you for the remark, it is an editorial error. In proper version, it should be written as: normal PT, thank you, we have recorrected this issue and checked the whole article content in terms of comparable mistakes.
Line 269: “We have…” I consider that this paragraph is more suitable for the conclusions section.
We have put the section to ensure the order of the analysis to ensure that further steps are more understandable and clear for the reader. Extended comments are placed in the Discussion and Conclusion sections.
Discussion
The discussion and conclusion were well described and provide new information that may help to differentiate between benign and malignant when PET/CT is used.
Thank you for your kind remark.
Overall comments:
This study was well made, the authors compared tonsillar lesions and results with the PET/CT. I think that authors provide interesting information about differential diagnosis using PET/CT (I, vs M). However, I have some questions that authors need to resolve.
- Why did not the authors use any photomicrographs of the carcinomas and lymphomas, as well as photomicrographs of the immunohistochemistry?
The reason why we did not include the photomicrographs in the article’s content is the overall volume of data, figures and tables. We did not want to confuse the reader with additional pages filled with images. We consulted the structure and number of pages with the Journal and we have been advised to make the initial version of the article shorter and to build the Supplementary Materials consisted of IHC and histopathology in comparison with the PET/CT results and comment them in the article’s content. So, all requested materials are presented in appropriate section according to the Journal recommendations.
- Why did the authors encompass only one group of carcinomas and lymphomas, when the histopathological features and behavior are different?
Because in terms of PET/CT diagnosis (images analysis) there are not metabolically different. They react similarly when using the DTP studies (more often increasing metabolic value over time). Only when the Lymphoma is highly advanced, it is more obvious. While locoregional - they might be confusing. We have studied general tendencies. Based on our previous experiences, we can analyze both SCC and Lymphomas in terms of SUVmax value cut-offs and now – differences and proportions as one group.
- I think it may be interesting to only contrast malignant lesions (SCC vs, DLBCL) and PET/CT. The results could be different when these pathologies are compared with inflammatory PT.
Thank you for the suggestion, we are planning to do so when the groups would be numerous enough and homogenous to the point, we can make the analysis reliable.
Reviewer 3 Report
This is an interesting paper but with some concerns.
- I found he english to be poor and needs a lot of work.
- What grades were the tumours? Really only T1N0 SCCs are difficult to diagnose.
- MRI is actually the first diagnostic method used and need to be mentioned.
- Oropharynx is the correct term for the tonsils, not mesopharynx.
- The results show promise but we need more clinical information, which may be impossible for the authors to provide. It is really only necessary to do PET when the diagnosis is in doubt. If clinically apparent or symptoms are "Red flag" then a biopsy is immediately performed, which is actually cheaper than PET in many countries and health care symptoms. This study needs to be restricted to patients in who a) the tumour was not obvious on clinical examination and b) the symptoms were "vague" enough to not merit immediate biopsy. I'd imagine the resuklts will be the same but in this cohort, these restrictions may underpower the study.
Author Response
We would like to sincerely thank the Reviewer #3 for all comments. We found them valuable and we have introduced corrections into the article’s content.
This is an interesting paper but with some concerns.
Thank you for the kind remark.
1. I found he english to be poor and needs a lot of work.
We have reviewed and recorrected the article.
2. What grades were the tumours? Really only T1N0 SCCs are difficult to diagnose.
Thank you for the remark, we did not clearly described the studied group. All diagnosed patients’ have been scanned with the contrast-enhanced computed tomography which showed suspicious cervical lymph nodes and no primary focus visible. Patients with CUP syndrome have been scanned with PET/CT to find the primary tumour. We have specified those in the Material and Methods section, accordingly with this remark. It is worth to mentioned that histological grading is a powerful tool for patients suspected follow-up, but it should be done only when it is possible full surgical removal – according tumor heterogeneity. In tumors of unknown primary it is hard to predict the appropriate grade scale. Additionally, lymphomas are graded in different way (usually into two groups: indolent lymphomas and high-grade lymphomas) than solid tumors (primary grading was developed into four-grade scale while in many places three-graded system is used.
3. MRI is actually the first diagnostic method used and need to be mentioned.
The idea of the first diagnostic method is in fact open for discussion and the Authors are not perfectly convinced which method should be used first. The presented work is just one set of data regarding this issue. Some publications indicate the MRI, other – 18F-FDG PET/CT.
Some of them suggest a PET/MRI. In our institution, the first diagnostic method to perform is the contrast-enhanced CT as a more available method of imaging when compared with the MRI and PET/CT. However, we have adjusted the article’s content in terms of this comment.
Lines: 59-60
4. Oropharynx is the correct term for the tonsils, not mesopharynx.
We have recorrected those. Thank you for the remark.
5. The results show promise but we need more clinical information which may be impossible for the authors to provide. It is really only necessary to do PET when the diagnosis is in doubt. If clinically apparent or symptoms are "Red flag" then a biopsy is immediately performed, which is actually cheaper than PET in many countries and health care symptoms. This study needs to be restricted to patients in who
a) the tumour was not obvious on clinical examination
b) the symptoms were "vague" enough to not merit immediate biopsy
I'd imagine the resuklts will be the same but in this cohort, these restrictions may underpower the study.
Thank you for the remark. We have attached to the submission the Supplementary Materials, which included more clinical data and comparison: markers – the PET/CT final conclusion. One of the goals of conducting these types of studies is to help the surgeon and pathologist to decide which area or region should undergo the biopsy. It seems to be impossible in practice to evaluate too many regions at once. We would like to help as much as we can to limit the surgical intervention which significantly affects patients. In this study, we have included patients in whom ceCT did not indicate the primary tumour, so the diagnosis has left some doubts. We have adjusted the paper and clarified this matter in the Methods section.
Round 2
Reviewer 1 Report
All comments have been addressed
Reviewer 2 Report
Dear authors, thank you, to respond me all cuestions and resolve my doubts.
I tink that your manuscript improved susbtantially and now I recommend this for publication.
Reviewer 3 Report
Thanks you for asking me to re-review this paper. I still have major concerns re the English quality throughout. With the small number of patients, the lack of TNM grading and limited use of MRI, I’m not sure I can find how this paper adds to the literature.